

# Talk the talk and walk the walk. Evaluation of autonomy in aging and Alzheimer disease by simulating instrumental activities of daily living: the S-IADL

Véronique Quaglino[1], Yannick Gounden[1], Emilie Lacot[1,2], Frédérique Couvillers[3], Amandine Lions[4] and Mathieu Hainselin[1]

[1] CRPCPO, EA, Université de Picardie Jules Verne, Amiens, France
[2] Service de Génétique Clinique et Oncogénétique, Centre Hospitalier Universitaire d'Amiens Picardie, Amiens, France
[3] Services de Neurologie et de Gérontologie, Centre Hospitalier Universitaire d'Amiens Picardie, Amiens, France
[4] IME Les Martinets, Adapei 36 l'Espoir, Saint Maur, France

## ABSTRACT

**Objective**. The autonomy of individuals is linked to the achievement of instrumental activities of daily living that require complex behavior. In the elderly, the assessment of autonomy is usually based on questionnaires that have strong subjective constraints. Considering this fact, we tested elderly healthy adults and Alzheimer disease patients using a new measure, the S-IADL (Simulation of Instrumental Activities for Daily Living), to assess the ability to perform effectively activities of daily living.

**Method**. The S-IADL shares many items with the well-known IADL questionnaire proposed by *Lawton & Brody (1969)*. However, as opposed to the IADL, the assessment of autonomy is not based on the completion of a questionnaire but requires the realization or simulation of various activities of daily living. Eighty-three participants (69 healthy elderly, and 14 Alzheimer Disease patients) completed the IADL and performed the S-IADL assessment.

**Results**. Results revealed that, like the IADL, the S-IADL is able to identify AD patients who are likely to encounter difficulties in performing everyday activities, and no major differences were found between the IADL and the S-IADL.

**Conclusions**. We outlined some advantages for prefering, in certain situation, this new tool based on simulation of activities in functional evaluation. Finally, we discuss the main limits of the S-IADL that should be investigated prior to its utilization by clinicians.

Corresponding author
Véronique Quaglino,
veronique.quaglino@u-picardie.fr

## INTRODUCTION

The present study concerns the estimation of personal autonomy in aging and Alzheimer disease. More precisely, instead of using typical self-evaluation questionnaires, we devised a new tool to evaluate individuals' capacity in performing common activities of daily living.

Two considerations guided this new tool proposal. First, the evaluation of autonomy must reflect to some extent, the real life of the person and secondly, it should not rely solely on the appreciation of the patient and caregivers who may not always be objective. We thus proposed a new tool which accesses autonomy through the simulation of daily living activities.

Autonomy is a systemic and multidimensional entity which encompasses sensorimotor, psychosocial, cognitive and medical aspects (*Eghbal-Téhérani & Makdessi, 2011*). Physical activity, social relationships, psychological well-being and cognitive functioning are thus among the many factors that can give an insight on an individual's degree of autonomy. The term functional autonomy refers to the ability to perform activities of daily living (ADL) (*Perrig-Chiello et al., 2006*), at different levels: basic activities (eating, moving, toileting or shower/bath taking) and more complex instrumental activities (medication taking, shopping or managing a budget). These instrumental activities of daily living (IADL) require good organizational, judgment and sequencing abilities (*Royall, Chiodo & Polk, 2000*). The loss of functional autonomy is generally the result of an imbalance between the functional capacities of an individual and the social and material resources available. Functional autonomy generally decreases with aging, as a consequence of the wide range of physical, cognitive, emotional, and/or social changes. This loss of autonomy is even more important for people with Alzheimer Disease (*Jekel et al., 2015*). However, these changes being characterized by interindividual heterogeneity, the assessment of elderly people's functional autonomy should therefore be specific and personalized in order to target their needs and to develop appropriate services. This kind of assessment is to our knowledge problematic due to a lack of appropriate tools for an efficient evaluation of functional autonomy in elderly adults.

In France, the AGGIR (*Autonomie Gérontologie Groupes Iso-Ressources* for Autonomy Gerontology Groups Iso-Resources) grid is commonly used to assess the degree of dependence of a person by classifying him or her, in one of the 6 categories (GIR 1 to GIR 6). Personal Autonomy Allowance (PAA) is granted to dependent older people classified in the first 4 GIR groups. People classified in GIR 1 are considered to be in the highest dependency and are thus allotted more help, while people classified in GIR 4 have the lowest help (*Bontout, Colin & Kerjosse, 2002*). In 2000, the Handicap-Incapacity-Dependency survey found that 530,000 elderlies out of the 12.1 millions of people aged over 60 years were classified in the GIR 1 category (*Colin & Coutton, 2000*). This data highlights the importance of developing specific tools for evaluating the autonomy of our elders who might be assigned to the same particular category while encountering their own specific difficulties with regard to autonomy.

The ADL scale (*Katz et al., 1970*) and the IADL questionnaire from *Lawton & Brody (1969)* assesses respectively the capacity of an individual to perform common ADL (body care, dressing, toileting, transferring, feeding...) and complex IADL (such as using the telephone, shopping, preparing meals, cleaning, washing clothes, using public transport, managing drug intake or a budget...). These tools are widely and frequently used in clinical practice and research (*Martyr & Clare, 2012*). In France, the IADL is even recommended by the National Authority for Health (HAS: Haute Autorité de Santé), an independent public scientific authority with an overall mission of contributing to the regulation of the

healthcare system by improving health quality and efficiency. Various studies have also confirmed its reliability and usefulness. For instance, the French PAQUID study (*Barberger-Gateau et al., 1999*; *Pérès et al., 2008*) showed that the degree of dependence as measured by the IADL self-report questionnaire, mostly with 4 items (ability to use telephone, mode of transport, responsibility for own medication, and ability to handle finances) could be a good predictor of the risk of developing dementia. More specifically, cognitive performances assessed by neuropsychological tests appear to be closely linked to the functional autonomy of individuals (*Perrig-Chiello et al., 2006*; *Tomaszewski Farias et al., 2009*). More recently, it was also found that attentional, memory, language, and visuospatial capacities correlated with the scores obtained at the IADL questionnaire specifically for telephone usage, drug taking and budget management (*Millán-Calenti et al., 2012*). However, these assessments are limited mostly to people aged under 80 years (*Jekel et al., 2015*).

Although the IADL has proved its efficacy, this evaluation of daily life autonomy suffers mainly from the absence of execution in real life context. Indeed, this self-assessment scale is presented in the form of a subjective self-report questionnaire and it requires the individual to be able to estimate his or her own functioning and abilities in using effective strategies when solving a particular task (*Juillerat van der Linden, 2008*). Metacognitive monitoring skills, which are demanding in terms of executive functions, are usually impaired in aging (*Douchemane, Isingrini & Souchay, 2007*; *Souchay et al., 2004*). These monitoring capacities are nevertheless particularly involved in various aspects of daily living activities (*Buckley et al., 2010*; *Mascherek et al., 2011*; *Mol et al., 2006*). For example, in Alzheimer disease (AD), including the mild stage, in addition to showing a lack of awareness with regard to their deficits (*Morris & Mograbi, 2013*; *Starkstein et al., 1997*) patients also have a tendency to underestimate their deficits in activities of daily living. It is worth noting that a meta-analysis showed no difference in the effect sizes between self-rated and informant-rated functional ability and executive function (*Martyr & Clare, 2012*).

Numerous alternatives to the IADL or to similar self/informant report questionnaires exist. For instance, a variety of performance-based measures have been devised over the past 25 years, such as the (Revised) Direct Assessment of Functional Status (DAFS; *McDougall et al., 2010*). These tools consist of presenting examinees with functional tasks in a standardized format. As an illustration, instead of questioning the patient on his or her cooking skills, the latter is required to prepare a meal using a mock kitchen within the hospital compound. However, this method of assessing functional autonomy is generally time consuming and may require a considerable financial investment on certain materials (for review see *Moore et al., 2007*). The Assessment of Motor and Process Skills (AMPS), for example, is one of the best known and widely used standardized assessments which measures the quality of a person's motor and process skill performance through therapist observation of everyday tasks. However, the use of this tool requires attending specific training workshops, but for many clinicians, time and money for training are continually reducing to the point of being nonexistent in some areas. Added to this, as noted by *Wenborn (2007)*, it is also difficult for many users to have their employers invest in the updated software of the AMPS. The potential range of barriers to effective implementation

of the AMPS have previously been well described by *Chard (2000)* and *Chard (2004)*. Finally, like most existing tools, the AMPS, to our knowledge, has not yet been implemented for a French speaking population.

Taking into consideration the French context, where the use of IADL is well established despite its various limits as outlined above (*Sikkes & Rotrou, 2014*) we proposed to create another complementary version of this tool. We took into consideration the main requests of clinicians: a tool which does not require lengthy training, not expensive to implement and that can be rapidly administered during a consultation.

We thus developed the S-IADL (Simulation of Instrumental Activities of Daily Living) which includes similar items to those of the IADL (*Lawton & Brody, 1969*). However, unlike the IADL, participants are required to simulate the activities and in doing so, this new tool aims to overcome the limitations of traditional assessments that we have outlined above. This study aims to match IADL and S-IADL performances for both healthy elderly (HE) participants and AD patients and to compare their sensitivity and specificity at their optimal cut-off points for the sample investigated.

## METHODS

### Participants

Eighty-three participants took part in this study: 69 healthy elderly (HE), and 14 Alzheimer Disease patients (AD). All participants were French native speakers, aged between 61 and 90 years old. They could all read, write, and understand correctly.

The 69 HE (including 37 women, 53.62%) were retired volunteers leading active lives. They were all seen at their own houses. HE with previous or current neurological or psychiatric disorders, sensory or motor impairments, and severe cardiac, respiratory or renal failure were excluded. The 14 AD patients (including 5 women, 35.71%) were recruited from day care centers. All of them previously underwent a complete diagnostic procedure conducted by a physician specialized in geriatrics and met the clinical criteria for probable AD according to the DSM-IV (*American Psychiatric Association, 1994*). The AD patients were seen in their day care center. They were all fully cooperative and none of them exhibited behavioral disturbances. The following criteria were also considered when selecting AD patients: apart from the AD, no other previous or current neurological or psychiatric disorders, sensory or motor impairments, severe cardiac, respiratory or renal failure. The Mini Mental State Examination (MMSE) (*Kalafat, Hugonot-Diener & Poitrenaud, 2003*), the Goldberg's anxiety Scale (*Goldberg et al., 1988*) and the Beck Depression Inventory (BDI–II) (*Beck et al., 1996*) were administered according to standard procedures to all participants. The characteristics of the two groups are shown in Table 1.

### Material

The Instrumental Activities of Daily Living questionnaire (IADL), translated in French by *Israel & Waintraub (1986)*, was used (*Lawton & Brody, 1969*). This questionnaire includes eight questions on people's abilities to use *Telephone*, to perform *Shopping*, to prepare *Meals*, to maintain *Cleaning*, to do *Laundry*, to use *Transport*, to take *Medications*, and to
**Table 1** Characteristics of the two groups of participants with *T*-Test and *p*-value comparisons.

| | Group | | *T*-test, *p* value |
| | AD (N = 14) | HE (N = 69) | |
|---|---|---|---|
| Age | 75–90 (M = 84) | 61–90 (M = 73) | – |
| MMSE | 23–18; M = 20.14 (1.99) | 30–22; M = 27.38 (2.05) | $T = 12.31, p < .001$ |
| BDI | 0–6; M = 2.21 (1.97) | 0–19; 4.16 (4.34) | $T = 1.63$, N. S. |
| Goldberg | 0–5; M = .79 (1.48) | 8–0; M = 2.04 (2.46) | $T = 1.84$, N. S. |
| IADL | 65–29; M = 49.76 (10.64) | 61–26; M = 30.9 (7.69) | $T = -6.31, p < .001$ |
| S-IADL | 51–8; M = 28.71 (13.49) | 61–0; M = 9.7 (9.67) | $T = -5.02, p < .001$ |

**Notes.**
Range score; Mean and standard deviation for the tests and comparisons between groups of participants (AD for Alzheimer Disease and HE for Healthy Elderly) using *T*-test and *p*-value.

manage *Budget*. The performance for each subtest of the IADL is estimated from 0 to 4. The total score is thus 32 points.

By referring to the IADL, we developed the S-IADL to simulate various instrumental daily living activities. The 13 activities of the S-IADL are likely to be representative of those performed by elderly adults (see Table 2 for details): planning a *Day;* making a *Shopping* list; opening *Hours* of shops; using the *Telephone;* paying a bill by *Check;* writing an address on an *Envelope;* planning how to relay point A to point B using a *Bus plan;* handling *Money;* putting away the *Shopping;* *Filling* a Pillbox; Taking *Medications;* managing *Accounts;* and *Household* activities. The performance for each subtest of the S-IADL is estimated from 0 to 4 points and the total score is thus 52 points.

To score, the examiner referred to an observation checklist to identify the strategies used, the difficulties encountered and also the strategies devised by the person to cope with these difficulties:

- *0 point*, unaided success: the action has been performed without any help or individual corrects himself or herself alone;
- *1 point*, partial success: the individual performs the action correctly when the examiner reported his or her error;
- *2 points*, help with success: the individual can achieve single action, but he/she succeeds with the help of the examiner;
- *3 points*, failure even with aid: the execution is unsuccessful even with the help of the examiner;
- *4 points*, failure: the execution is impossible or not evaluable.

For the two questionnaires, IADL and S-IADL, the higher the score is, the less the individual is autonomous.

## Design

The factors considered in this study corresponded to a 2 (group: HE and AD) × 2 (evaluation: IADL and S-IADL). The first factor was varied among participants and the second one was a within-participants factor.

**Table 2  S-IADL items and their description.**

| S-IADL items | | IADL items |
| --- | --- | --- |
| 1–Planning a day | Organize a typical day by matching activities written on labels with a particular time of the day. This item is composed of a board with schedules for a day and 10 labels. | / |
| 2–Making a shopping list | Make a list of groceries to buy for a determined number of relatives in order to be able to eat over two days. | Shopping |
| 3–Opening hours of shops | Verify, by referring to three flyers of different shops, whether they are open on Monday at 10:00 o'clock. | Shopping |
| 4–Using the telephone | Make a phone call to ask for the opening hours of the butcher (number in a flyer). | Ability to use telephone |
| 5–Paying a bill by check | Write a check to pay a phone bill. | Ability to handle finances |
| 6–Writing an address on an envelope | Put the check in an envelope, write the address and make a cross at the location provided for the stamp. | / |
| 7–Planning how to travel from point A to point B using a bus plan | Be at the post office at 10 o'clock to post the envelope containing the bill. A city map with two bus routes and corresponding schedules is presented. A red dot indicates on the plan where the participant resides. The task consists in choosing the right bus route and the appropriate time. | Mode of transportation |
| 8–Handling money | Give money (8–60) to the baker, using a purse containing different coins and bank notes. | Ability to handle finances |
| 9–Putting away the shopping | Classify various items according to three possible storage places (refrigerator, kitchen cupboard & table). Three boards with the name of the different storage location and their corresponding images are presented to the participant, together with a list of various foods and objects. | Shopping food preparation |
| 10–Filling a pillbox | Using a prescription, the participant is required to fill a pillbox with fake drugs for the week. | Responsibility for own medications |
| 11–Taking medications | By referring to the pillbox, the participant is asked three times what drugs must be taken at a given day and a specific time. | Responsibility for own medications |
| 12–Managing accounts | Taking into consideration a cashback ticket, invoices and receipts, the participant is asked to calculate how much money is left in his/her account. A calculator is available for facilitating calculation. | Ability to handle finances |
| 13–Household activities | The participant is asked to determine what household activity relates to the pictures shown. This item consists of two boards depicting 3 types of household chores (cleaning, doing laundry and washing dishes). The participant is also required to match 16 pictures of equipment or objects with these activities. | Housekeeping laundry |

## Procedure

The study was conducted in accordance with the ethical guidelines for research in psychology and with the ethical principles of psychologists and code of conduct. The HE adults were contacted by phone and after a brief explanation of the theme of the study, an appointment was set up. Each participant was then seen at home by a clinical psychologist. With the aid of an anamnestic questionnaire, a meticulous screening of the HE adults' general medical history was then performed. Concerning the AD patients, they were seen individually at their day care centers by the same clinical psychologist as for the HE adults. In order to enable participants to make an informed decision as to whether or not they wish to participate, they were asked to read and sign a consent form which stated the purpose

**Table 3** The Area under the ROC curve (AUC) for IADL and S-IADL questionnaires.

| | AUC | SE | 95% CI | | $p$ |
| | | | Min | Max | |
|---|---|---|---|---|---|
| IADL | .92 | .038 | .84 | .99 | $p < .001$ |
| S-IADL | .91 | .039 | .83 | .98 | $p < .001$ |

**Notes.**
SE, Standard error; CI, Confidence interval; $p$, Significance level.

of the study, what is expected from them, the length of time, the possibility of withdrawal at any time and also the guarantee of confidentiality and anonymity of personal data. The study was carried out according to the local Ethics Committee for Health Research.

Each participant was assessed in the same order with the following evaluations: The Mini Mental State Examination (MMSE, *Kalafat, Hugonot-Diener & Poitrenaud, 2003*), the Goldberg's anxiety Scale (*Goldberg et al., 1988*) and the Beck Depression Inventory (BDI–II, *Beck et al., 1996*). The IADL was then administered as a self-assessment scale, followed by the S-IADL. The clinical psychologist paid attention to the strategies used by the participants, the difficulties encountered and the palliative solutions that they deployed to overcome these difficulties. Overall, the experimental procedure lasted for about 1h30 and participants were given the opportunity to take a break when needed.

## RESULTS

Descriptive data (means proportions and standard deviation of the IADL and S-IADL scores) are presented in Table 1 and were compared using a Student's 2-sample $t$-tests with 5% as error rate level. Each total score was transformed into a success rate (percentage) Note that the higher the score is, the less the individual is autonomous. As expected, HE obtained better performances at both tools (IADL $M = 30.9$; S-IADL $M = 9.7$) than AD patients (IADL $M = 49.77$; S-IADL $M = 28.71$).

The sample of HE adults were younger than the AD patients, so an ANCOVA was conducted to control the confounding age factor (Table 3 and Appendix S1). Results showed a significant difference in IADL score between the two groups when adjusted for age [$F(1, 82) = 28.52, p < .001, \eta2 = .26$]. Regarding the estimation of the effect size, the partial eta-square coefficient is large explaining 26 % of the variance (*Cohen, 1988*). For the S-IADL scores, the ANCOVA revealed that there was also significant difference between the two participant groups when adjusted for age [$F(1, 82) = 13.31, p < .001, \eta2 = .14$]. Concerning the estimation of the effect size, the partial eta-square coefficient is large explaining 14% of the variance.

A Receiver Operating Characteristic (ROC) curve analysis was performed to estimate the optimal cut-off value for both questionnaires and to determine the ability of each to discriminate between groups AD patients and HE adults (see Fig. 1). We also compared the performances at both IADL and the S-IADL questionnaires by evaluating the areas under the curve (AUC). The AUC of the questionnaire was significantly different from the AUC corresponding to a random test. In order to determine their specificity and sensitivity for establishing their cut-off points, we searched for significant differences between the AUCs.

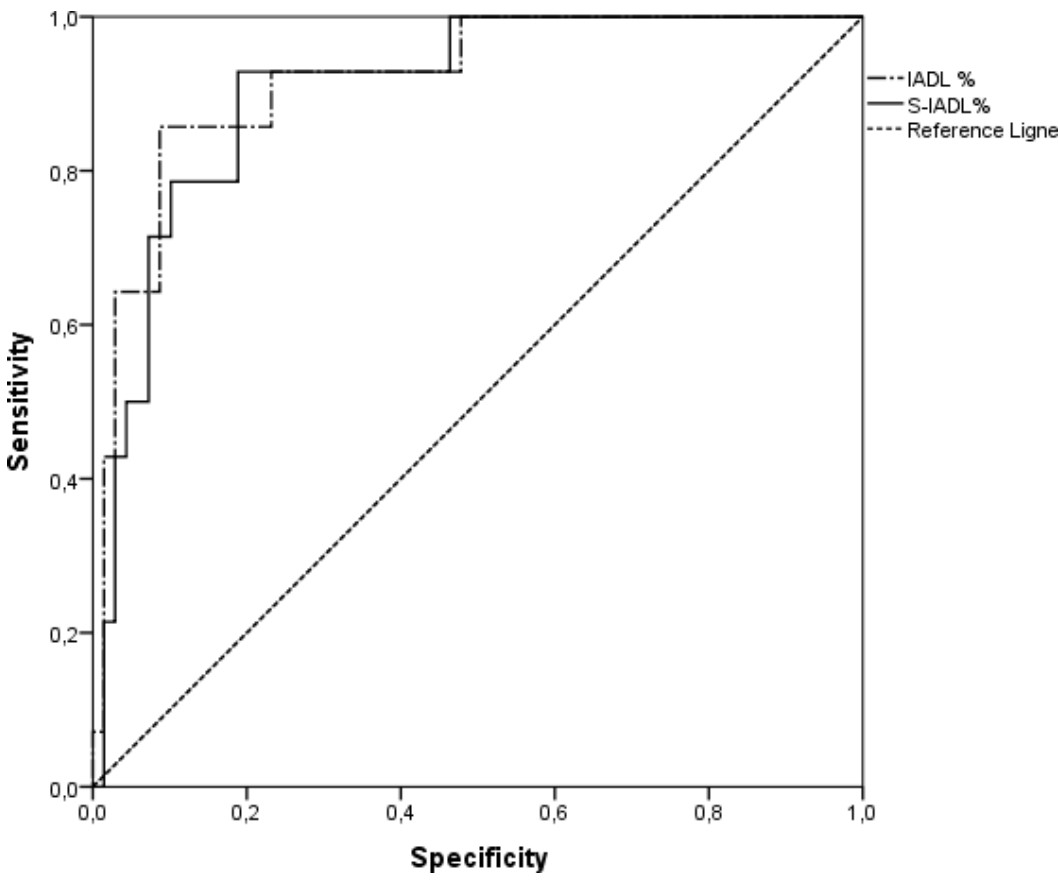

**Figure 1  ROC curve.**

We ended our analysis by comparing the sensitivity and specificity of the two questionnaires with their established cut-off points. In analyzing our results, we first checked whether the AUC of the two questionnaires was significantly different from the area under the diagonal determined of a random test. Table 3 presents the results of this analysis for the IADL and S-IADL questionnaires. As shown in the table, the test revealed a significant difference from the random area, with a probability of error smaller than 0.1 % ($p < .001$).

In order to establish the optimal cut-off point of each scale, we analyzed both sensitivity and specificity at each possible cut-off point. The best performance of the IADL in discriminating between HE and AD participants, is reached at the cut-off point of 41.94 (sensitivity = 86; specificity = 91). Note that this cut-off point has to be considered as the success rate that allows dissociating between a deficiency and a non-deficiency performance. For the S-IADL, results indicated that the best discriminating performance is reached at the cut-off point of 14.98 (sensitivity = 93; specificity = 81).

## DISCUSSION

The IADL is a self-assessment scale and is presented in the form of a subjective self-report questionnaire while the S-IADL requires an active participation to simulate the same activities as the IADL.

As expected, HE participants were better than AD patients in both the S-IADL and the IADL, even when age (as covariate factor) is adjusted. However, contrary to our expectations, this pattern of results was not significantly different across the two questionnaires. Our results also revealed that both the IADL and S-IADL questionnaires are able to identify AD patients who are likely to encounter difficulties in performing everyday activities. A high discriminative capacity was also found for the two questionnaires. It suggests that for both HE adults and AD patients, responses at the IADL and performances at the S-IADL questionnaires were significantly different from what could be obtained at a random test. These two questionnaires have thus a high accuracy in identifying AD patients who might be losing autonomy in everyday activities.

By referring to the accuracy of both questionnaires at their optimum cut-off point, it appeared that the S-IADL has a better sensitivity (93 vs. 86) than the IADL while the IADL has a better specificity than the S-IADL (91 vs. 81). However, it is difficult at this point to draw any conclusion based on these minimal differences. Further research is needed to confirm results for this preliminary study. If this pattern of results with regard to sensitivity and specificity is real, this would mean that the S-IADL and the IADL are complementary tools which could be preferred depending on the clinician's objectives.

The S-IADL is another step towards a more ecological way to assess patients' abilities and autonomy in a classical context of evaluation (i.e., without going to the person's home). Although in the present study no major differences were obtained between the classical IADL and the S-IADL, in certain situations it may be worth preferring our new tool. For instance, it may give clinicians the ability to identify why an activity is difficult to perform, how the person copes with his or her difficulties in order to perform the activity, and what can be proposed as specific compensatory strategies.

Poor self-esteem or lack of (or at least not always full) awareness is known to affect cognitive assessment (*Agrigoroaei & Lachman, 2011*; *Soederberg Miller & Lachman, 2000*) and can have considerable impacts on daily activities. In the first case, due to low self-esteem or lack of self-confidence, some ageing adults may feel that they are no more able to perform on their own certain daily living activities that they actually can. Inversely in the second case, not knowing that a deficit exists may make problematic for people to accept assistance (*Cott & Tierney, 2013*). Consequently, it is not rare to have a patient exposed to serious dangers by performing at home activities that should not be performed alone. However, it should be noted that a few studies have also shown that people with dementia may be aware of their own IADL ability (*Kiyak, Teri & Borson, 1994*; *Marková et al., 2014*; *Martyr, Nelis & Clare, 2014*). Awareness literature often focused on discrepancy scores between self and informant ratings, but failed to take into account all of the confounding variables influencing the reliability of the informant. For instance; recent studies show that caregiver burden, informant depression, age and cognitive status of the patient, are among the many factors associated with rating bias (*Martyr, Nelis & Clare, 2014*; *Sikkes & Rotrou, 2014*).

With regard to these two issues, the S-IADL can be an appropriate tool since it does not rely on subjective evaluations of patients and informants but rather places patients in an active position where they can be exposed to what really they can do or not. In the present study, no measure of awareness or self-esteem, was conducted. However, considering the

coherence between the IADL and the S-ADL, it is possible that overall, our participants have a good awareness of their actual capacities in daily living activities. Indeed, the AD patients were all at an early stage and did not present a serious deterioration of their global intellectual capacities (see MMSE score).

In the very early stages of AD, a combination of the IADL and the S-IADL might be useful especially when patients have low self-esteem with regard to their abilities in daily life activities. In later stages particularly when patients suffer from acute lack of awareness, the S-IADL may be more appropriate in assessing autonomy in everyday life activities.

However, further studies need to examine the relations between the degree of awareness or self-esteem with performances at both the IADL (what I think I can do) and the S-IADL (what I can actually do).

By using a specific grid, the S-IADL allows clinicians to gather more and multimodal information while patients are performing the actions. They can have a more precise idea of which specific step in the process is a problem and identify strategies which may be effective or not. All this information, unavailable with questionnaires, should help clinicians not only for diagnosis purposes but also for rehabilitation.

Another strength of the S-IADL is the ability to adjust items to a particular patient considering his/her actual activities. Besides the 13 activities described here, we can imagine adding gardening or sewing or any other daily activity that the person performs. This flexibility of the S-IADL is in line with the need to develop personalised care and support for patients.

Naturally, there are also some limitations to the S-IADL which must be resolved before proposing it to clinicians. Beside the need to test the S-IADL on a bigger sample, as suggested above we also need to include patients with and without deficit awareness and see whether performances would be different between the IADL and the S-IADL. Most importantly, it will also be necessary to verify that performances of a person with the S-IADL really reflect his or her abilities in real life. Indeed, one can have difficulty with daily living activities for various reasons such as cognitive and or motor impairments. Success or failure in the S-IADL is more likely to rely on the efficiency of cognitive functions and only to some extent on motor abilities, since patients are requested only to simulate the activities. Thus, although confirmation is needed, we can expect that someone who encounters difficulties in performing the S-IADL will also encounter difficulties in cognitively more complex real life situations. However, we will be more prudent in suggesting that a person who succeeds in the S-IADL is also able to perform with the same ease daily living activities in real life. Hence, like the various tests at the disposal of clinicians, the S-IADL is only a tool which can give some important indications on one's autonomy in performing daily living activities but is not sufficient on its own. It can be used as a complement to the IADL and a thorough interview of the patient and his or her relatives.

The IADL scale has been widely criticized for being strongly biased towards men since many of the items are based around traditional female gender roles. This potential gender bias of the Lawton IADL Scale has led to the generation of some non-validated adaptations for its application in men (*Vergara et al., 2012*). Investigating a possible gender effect with

the S-IADL is thus of utmost importance prior to its utilization by clinicians. Finally, although the IADL is known to have reasonably good cross-cultural validity (*Ng et al., 2006*), further research should also test the cross-cultural applicability of the simulation version across ethnic groups and if possible in conjunction with gender.

## ACKNOWLEDGEMENTS

We would like to thank all participants. Also we are grateful to Camille Gaudet, Marine Walton, Alexandra Benoist and Sophia Benquet, students in Neuropsychology at Picardie Jules Verne University (Amiens, France) who contributed to data collection.

### Funding
The authors received no funding for this work. Publication fees were supported by the CRPCPO laboratory. The funders had no role in study design, data collection and analysis, decision to publish, or preparation of the manuscript.

### Grant Disclosures
The following grant information was disclosed by the authors:
CRPCPO laboratory.

### Competing Interests
The authors declare there are no competing interests.

### Author Contributions
- Véronique Quaglino conceived and designed the experiments, analyzed the data, contributed reagents/materials/analysis tools, wrote the paper, prepared figures and/or tables, reviewed drafts of the paper.
- Yannick Gounden and Mathieu Hainselin analyzed the data, contributed reagents/materials/analysis tools, wrote the paper, prepared figures and/or tables, reviewed drafts of the paper.
- Emilie Lacot analyzed the data, contributed reagents/materials/analysis tools, prepared figures and/or tables, reviewed drafts of the paper.
- Frédérique Couvillers performed the experiments, analyzed the data, contributed reagents/materials/analysis tools, reviewed drafts of the paper.
- Amandine Lions performed the experiments, contributed reagents/materials/analysis tools, reviewed drafts of the paper.

### Human Ethics
The following information was supplied relating to ethical approvals (i.e., approving body and any reference numbers):

All participants could read, write, and understand correctly and were fully informed about the aim of the study and procedure details. They signed an informed consent form which assured them that their answers would remain confidential and anonymous. The study was carried out according to the local Ethics Committee for Health Research.

## Data Availability

The raw data has been supplied as a Supplementary File.

## Supplemental Information

Supplemental information for this article can be found online at http://dx.doi.org/10.7717/peerj.2351#supplemental-information.

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
