# Peer review of "Talk the talk and walk the walk. Evaluation of autonomy in aging and Alzheimer disease by simulating instrumental activities of daily living: the S-IADL"

_PeerJ, doi:10.7717/peerj.2351_

## Round 0.1 · original submission · Major Revisions

Dear Authors,

Four peer reviewer have given their comments and after reading the manuscript and their comments a major revision decision is given because I believe that Peer Reviewer 1 has points that I hope your can seriously look into to make your manuscript better.

Please resubmit the revised manuscript once it is ready.

Thank You.

·

Basic reporting

The authors present a paper that looks at evaluating a new test of functional ability. The paper is adequately written though suffers from numerous typos, the authors do not seem to have reviewed the extant literature particularly thoroughly with no reference to previously devised tests that measure similar activities. The main comparison between the two groups is flawed due to the differences in age of the two groups and reanalysis is necessary.

The range of test scores should also be included for all measures; this is especially important since the authors refer to sensitivity cut offs.

There are numerous typos and the manuscript would benefit from being read by a native English speaker.

Experimental design

A major limitation of the paper (and it’s not clear why the authors didn’t control for this) is that the healthy older people sample are considerably younger than the dementia sample (mean ages 73 vs 84, respectively). Based on this difference it is not surprising that the AD group were more impaired since age has been found to be associated with functional ability; the authors should conduct ANCOVA controlling for age. A further limitation of the study is that most of the AD group were men, the Lawton & Brody IADL scale has been widely criticised for being strongly biased towards men since many of the items are based around traditional female gender roles and certainly in the age groups under investigation in the current study those activities were mostly performed by women, thus men are rated as more impaired based purely on the fact that men rarely, or never undertook some of the tasks in the questionnaire.

Validity of the findings

The authors argue that people with dementia lack awareness of their own functional ability however, there are a few studies that have shown that people with dementia may actually be aware of their own IADL ability (Kayak et al, 1994, Martyr, Nelis & Clare, 2014). Most of the awareness literature is based on discrepancy scores with the majority of studies suggesting that a difference between self-ratings and informant ratings indicates evidence of a lack of awareness, however there are many confounding variables that too many studies ignore such as carer burden (Mangone et al., 1993; Martyr, Nelis, & Clare, 2014; Razani et al., 2007; Slachevsky et al., 2013 and many others), the age of the person with dementia (Martyr et al, 2012, Kayak et al, 1994), the cognitive status of the person with dementia (Martyr et al., 2012; Teri, Borson, Kiyak, & Yamagishi, 1989), which have all been found to contribute to a bias in informant ratings. It is therefore problematic to say that people lack awareness of functional ability. Indeed, the findings of the current study show that people with AD rated their own functional ability as more impaired than the controls suggesting some level of awareness of their own functional difficulties so it is not clear why the authors talk about a lack of awareness rather than talk about the level of awareness the AD group display.

The S-IADL as described in the paper shares a number of similar tasks with many performance based tests of functional ability such as the Direct Assessment of Functional Status or the Assessment of Motor and Process Skills (AMPS). The AMPS for example is primarily used by occupational therapists and has a large number of activities to choose from, so consequently has vast scope for assessments of personalised care. Could the authors say how the S-IADL differs from these earlier tests and what it adds to the already large number of functional assessments? The authors may find the reviews by Moore, Palmer, Patterson, & Jeste, (2007), Martyr & Clare, 2012, and Sikkes & Rotrou, (2014) helpful.

·

Basic reporting

Basic reporting is good , organized and well elaborated. specifically there was good consideration on ethical aspects.
Research question were missing in reporting.

Experimental design

which experiment design was used, it is not clear, the method of sampling and sampling distribution need to be elaborate more and need to justify the division of participant.
Explain about the Stimulation activities.

Validity of the findings

findings are helpful but need more research study to validate the newly established method of S-IADL.

Comments for the author

Overall it is helpful study to evaluate stimulating instrumental activities of daily living among aged people.

Reviewer 3 ·

Basic reporting

The manuscript has an established flow and sufficient English grammatical language. The introduction and justification to the need of the paper is highlighted.

Experimental design

The method involved in the study design and analysis are justified. The raw data attached acts as a proof of data collation and management.
Statistical analysis is acceptable and has been discussed on findings.

Validity of the findings

Findings are explained and has been pointed to their specific sources. Inclusion and control of data recruited have been explained.
Please add and stress on the further use and suitability of the tool for health and unhealthy elders and Alzheimers as these two conditions may project different conclusions if wrongly used.

Comments for the author

Please comment on the use of these two tools in other countries such as low or middle income countries.

Reviewer 4 ·

Basic reporting

The article is well written and easy to understand the main message the authors want to communicate.

Experimental design

1. The authors clearly described the design of the study and the steps involved.
2. The research question is relevant, meaningful and important; especially to aging societies.
3. The investigation was well conducted.
4. Enough information was provided by the authors to understand the methods applied herein
5. Ethical standards were followed in conducting the study.

Validity of the findings

1. I have no comments on the data. however, the statistical analysis provided in the paper was appropriate.
2. There is no information on data access. I hope the authors will provide more information on that.
3. The conclusion from the paper was appropriate; however, I would like the authors to elaborate a bit more on the finding that " (see line 260: contrary to our expectations, the pattern of results was not significantly different across the two questionnaires (IADL and S-IADL). If the S-IADL questionnaire (proposed by the authors) is not better/or different from the conventional IADL questionnaire, under what conditions will S-IADL be preferred to the IADL questionnaire.
4. The study was well described and could be replicated in another setting

Comments for the author

1. could the authors elaborate on the benefit of using S-IADL to the IADL questionnaire (clinical or policy or social )
2. There was very little information on the motivation for the study. The authors referred to the fact that IADL questionnaires are subjective. This I believe could be the case for S-IADL. Because knowing how activities of daily living are done is different from being able to do it. could the authors speak to that?
3. The study used only one examiner to observe the participant of the study simulate the IADL? Is there a possibility of examiner bias ? would multiple examiners reduce the likely examiner bias?
4. Any explanation for the difference in sensitivity and specificity of S-IADL and IADL
5. I will like to see table 2 rearranged to match S-IADL items to IADL items.

---

## Round 0.2 · Minor Revisions

Dear Authors,

Please heed the advice of Reviewer 1 who has provided additional which I agree are important. Also note their Annotated manuscript

Thanking you

·

Basic reporting

The authors have addressed many of the comments that were suggested previously and the authors should be commended for doing a good revision. I have a few minor suggestions.

Experimental design

In the method section I was confused by the total scores and the data presented in Table 1. The iADL was said to have a total score of 32 and the S-iADL had a total score of 52. Yet for the iADL the mean AD score was 49.76 and the ranges went from 29-61 and for the HE group 26-65. Similarly, the range for the S-iADL the HE group had the highest score of 61. I may have missed where this disparity was explained in the method section.

Validity of the findings

In the results section there are no statistics for the covariates, there should be data for age in the ANCOVA and this should be included in the paper. Could the authors explain how an n2 score of .14 is “large”. Could you provide a reference for any cutoffs that you used to help interpretation?

Comments for the author

In the introduction the authors state that “Moreover, it is also found that patients’ cognitive abilities correlate more with information transmitted by caregivers than those given by the patients themselves.” However, the Martyr & Clare, 2012 meta-analysis found the opposite of this since there was no difference in the effect sizes between self-rated and informant-rated functional ability and executive function. It should also be noted that this meta-analysis found that the iADL was the most frequently employed questionnaire of IADL.

Also in the results section the authors state that “We ended our analysis by comparing the sensitivity and specificity of the two questionnaires with their previously established cut-off points.” These cutoffs were not mentioned previously.

The discussion states that “The S-IADL is a first step towards a more ecological way to assess patients’ abilities” this is not a first step, there have been many studies that have employed simulated assessments of functional ability. As stated in my previous review the DAFS has been used the most frequently (Martyr & Clare, 2012) in dementia and there is also an adaptation of the DAFS that has been used with healthy older people (McDougall et al, 2010; The Gerontologist). This McDougall adaptation is very similar in content to the S-iADL so should be mentioned.

The manuscript still has a number of typos and some grammatical oddities.

·

Basic reporting

Basic reporting is up to standard.

Experimental design

experimental design is improved and and sample is justified

Validity of the findings

Study findings are good to lead further research in cognitive areas of functioning to improve daily living skill with cognitive impairment in different cases.

Comments for the author

over all study is good, and suggested instrument of S-IADL need more experiments to be validated.

Reviewer 4 ·

Basic reporting

I'm satisfied with the revision made by the authors

Experimental design

I'm satisfied with the revision made by the authors

Validity of the findings

I'm satisfied with the revision made by the authors

Comments for the author

I'm satisfied with the revision made by the authors

---

## Round 0.3 · accepted · Accept

Dear Authors,Thank you for your revised manuscript which has been accepted for publication.

Please add the missing data to Table 1

·

Basic reporting

The authors have addressed the comments that were suggested previously and the authors should be commended for doing a good revision.

Experimental design

The design is clearly described and the additional information that the authors have supplied helps interpretation.

Validity of the findings

The findings are sound. The additional ANCOVA analysis included in the appendix is interesting and suggests that the Lawton’s IADL measure has a significant gender effect (though the effect size is small) whereas the SIADL does not have a gender effect, this could have been mentioned in the paper. The effect of age in both the subjective and objective measure is interesting and suggests that as people age their ability to perform everyday activities decreases. I also noticed that there is some missing data in Table 1 for the Becks and Goldberg measures in the AD group, this will need to be added to Table 1.

Comments for the author

The paper is much improved and I am satisfied with the revisions made by the authors.